# *Nannochloropsis* spp. as Feed Additive for the Pacific White Shrimp: Effect on Midgut Microbiology, Thermal Shock Resistance and Immunology

**DOI:** 10.3390/ani11010150

**Published:** 2021-01-11

**Authors:** Ariane Martins Guimarães, Cristhiane Guertler, Gabriella do Vale Pereira, Jaqueline da Rosa Coelho, Priscila Costa Rezende, Renata Oselame Nóbrega, Felipe do Nascimento Vieira

**Affiliations:** 1Laboratório de Camarões Marinhos, Universidade Federal de Santa Catarina, Florianópolis 88061-600, Santa Catarina, Brazil; arianeufsc@gmail.com (A.M.G.); jaquesombrio@gmail.com (J.d.R.C.); priscila.pesca.ufal@hotmail.com (P.C.R.); 2Campus São Bento do Sul, Instituto Federal Catarinense–São Bento do Sul, São Bento do Sul 89283-064, Santa Catarina, Brazil; cristhiane.guertler@ifc.edu.br; 3Sparos I&D–Nutrition in Aquaculture, Marim, 8700-221 Olhão, Portugal; gabriellapereira@sparos.pt; 4Laboratório de Nutrição de Espécies Aquícolas, Universidade Federal de Santa Catarina, Florianópolis 88066-260, Santa Catarina, Brazil; renata.oselamenobrega@gmail.com

**Keywords:** *Litopenaeus vannamei*, EPA, low temperature thermal shock, immunological parameters, reactive oxygen species

## Abstract

**Simple Summary:**

We evaluated *Nannochloropsis* spp. as a feed additive for the Pacific white shrimp by testing for thermal shock resistance, immunology, and midgut microbiology. *Nannochlorpsis* spp. are rich in Eicosapentaenoic acid, an n-3 fatty acid. This study demonstrated that the inclusion of these microalgae in the diet increased shrimp resistance to thermal shock and stimulated shrimp immune defense.

**Abstract:**

This work aimed to evaluate *Nannochloropsis* spp. as feed additive in the diet of Pacific white shrimp for their effect on midgut microbiology, thermal shock resistance and immunological parameters. Initially, the digestibility of the microalgae meal was assessed, and the apparent digestibility coefficient (ADC) was determined. The ADC was, in general, high in lipids (78.88%) and eicosapentaenoic fatty acid (73.86%). Then, *Nannochloropsis* spp. were included in diets at four levels (0, 0.5, 1 and 2% inclusion). The shrimp were reared in 500 L clear water tanks containing 20 shrimp per tank with an initial weight of 6.05 ± 0.06 g and fed four times a day. Shrimp fed with supplemented diets containing *Nannochloropsis* spp. (0.5 and 2%) presented higher resistance to thermal shock when compared to the non-supplemented group (control). Shrimp fed with 1 and 2% of algae inclusion had a higher production of reactive oxygen species (ROS) when compared to other treatments. No statistical difference was observed in the immunological parameters and microbiology of the intestinal tract. Thus, the inclusion of *Nannochloropsis* spp. in shrimp diets at 0.5 and 2% levels increases resistance to thermal shock and ROS production in shrimp.

## 1. Introduction

During the past few years, worldwide shrimp production has encountered limited development and production owing to, among other factors, viral and bacterial outbreaks [1]. Many methods have been used to treat or prevent the development of new diseases, such as chemical and antibiotic compounds. However, the harmful potential effect of synthetic drugs on animals and the environment made these compounds problematic. Thus, to avoid their use, alternative methods have been studied, for instance, good handling practices of production and the enhancement of the immunological resistance of shrimps. The balance among host, pathogen and the environment is crucial for the animals that, even though they come into contact with the infection agent, do not develop the disease. Avoiding fluctuations in salinity and temperature are also essential practices in shrimp production, since they are stressful factors and can eventually serve as a stimulus for the development of an outbreak. On the other hand, the development of the immunocompetence of shrimps strengthens their immune system. This can be developed throughout the use of natural substances in diets, namely immunostimulants, such as some plant additives and seaweeds [2,3,4,5,6].

Immunostimulant composts can be synthetically produced or obtained through natural sources, for instance through ingestion of microorganisms, such as fungi and bacteria, as well as algae, or even microalgae and seaweed [7,8]. Microalgae are already part of the food chain of marine shrimp. They are a source of long-chain fatty acid and are extremely important for penaeid shrimp, since their capacity to synthetize those fatty acids is limited [9,10,11,12,13,14]. Fatty acids are part of the group of lipids and are important because they are components of cell membranes and a source of energy. Studies with immunostimulants have demonstrated their positive effects on the growth and modulation of the immune system of Pacific white shrimp [15,16,17].

Microalgae from the *Nannochloropsis* genus are well known in aquaculture nutrition and play an important role in live feed enrichment because they are rich in lipids and contain high levels of eicosapentaenoic fatty acid. In addition, some species are also rich in carotenoids, such as *N. salina* and *N. oculata* [18,19,20]. The benefits of *Nannochloropsis* spp. in shrimp feeding are already well-studied, and they range from improving growth performance and survival to enhancing fatty acid content in shrimp flesh [21,22,23]. These microalgae contain β 1-3 glycans in their cellulose cell wall. After analyzing some important immunological parameters for Pacific white shrimp, some studies have suggested that this cell wall component is responsible for shrimp immunostimulation [24,25,26,27,28,29,30].

Immunological parameters such as hemograms, hemolymph coagulation time, phenoloxidase (PO) activity, the agglutination titer of plasma, plasma total protein concentration, and reactive oxygen species production (ROS), are used to verify the health of crustaceans. ROS is a common measurement of immunostimulation in shrimp [24,25,26,27,28,30,31,32,33]. Among all ROS, superoxide anions are especially active against pathogens, owing to the occurrence of oxygen consumption, known as respiratory “burst”, during the production of these molecules, which inhibits growth and finally destroys the intruding pathogen [1,34].

This work aimed to evaluate *Nannochloropsis* spp. as a feed additive in the diet of Pacific white shrimp feeds for their effect on midgut microbiology, thermal shock resistance, and immune defense.

## 2. Materials and Methods

Experiments were performed in the Marine Shrimp Laboratory (LCM), Federal University of Santa Catarina (UFSC), Barra da Lagoa, Florianópolis (SC). *Litopenaeus vannamei* from the lineage high health SPEEDLINE Aqua were bought from Aquatec Aquacultura Ltd.a. (Canguaretama, RN, Brazil) and cultivated in LCM facilities in a biofloc system until reaching the correct weight. 

### 2.1. Experimental Diets

Microalgae meal from *Nannochloropsis* spp. was purchased at Necton—PhytoBloom (Algarve, Portugal) (Table 1). Initially, two diets were formulated for the digestibility trial and another diet was formulated for the feed additive inclusion trial. Both diets were formulated with the Optimal Fórmula 2000 software, version 2009/2010, and the nutritional requirements were based on the species (*Litopenaeus vannamei*). For requirements not specified for *L. vannamei*, the specie *Penaeus monodon* was used as reference [35]. The ingredients’ proximal compositions uploaded into the software were based on reports shared by the supplier and research reports in the literature.

To perform the digestibility trial, a reference diet was formulated using only the semi-purified ingredients (Table 2). Yttrium oxide 1.00 mg kg^−1^ was added as an inert marker. The test diets contained 900 mg g^−1^ of the dry weight of the reference diet and 100 mg g^−1^ of *Nannochloropsis* spp. meal. Diets were formulated to have 40% crude protein, 3762.07 kcal k^−1^ of energy and 7.48% lipids. The ingredients were crushed and sieved in mesh of 600 µm, and then mixed in the following sequence: macro ingredients and then micro ingredients. After each mixture, oils and soy lecithin were added. Finally, the humidity was adjusted to 15%. The diets were pelletized at 1.5 mm diameter and dried at 40 °C for 12 h. Both diets were kept frozen until the time of feeding to avoid fatty acid loss.

For the experiment with additives, diets were formulated to contain 35.86% crude protein, 3762.07 kcal kg^−1^ of energy and 10% lipids (Table 3). The ingredients were previously ground and sieved in a 600 µm mesh. Then, all macro ingredients were mixed, and, later, the micronutrients were added. The master mix was separated into 4 equal parts, and the microalgae meal was added at 0, 0.5, 1, and 2% from the total volume of each part. After that, each mix was pelleted, dried, and frozen, as described above, for the digestibility diet.

### 2.2. Experimental Design and Conditions

#### 2.2.1. Digestibility Assay

For the digestibility assay, a total of 400 shrimp were used (8.00 ± 0.30 g). The shrimp were stored in two of 5000 L clear marine water tanks and fed four times a day (8 h, 12 h, 14 h and 18 h), followed by acclimation for 10 days with the diets. After this period, 10 shrimp from each treatment were transferred to a 60 L tank. Feces collection was made in quadruplicate for each treatment resulting in a total of 8 experimental units in a room fed by constant aeration (5.58 ± 0.44 mg L^−1^) and heating (28.17 ± 0.51 °C).

The animals were fed twice a day (8 a.m. and 2 p.m.) and after each feeding, the tank was siphoned to retirated the feces and feed remaining in the tanks after 40 min of the end of feeding. During the following 4 h, the feces collection started with the use of Pasteur-type pipets. The feces were immediately transferred to Petri dishes, washed with distilled water, transferred to a Falcon tube, and immediately stored in ice to avoid the degradation of fatty acids. At the end of the day, tank water was renewed 90%, and the Falcon tubes were centrifuged twice at 1800× *g* at 4 °C for 10 min (model 5804R Eppendorf AG, Hamburg, Germany) for excess water removal and then frozen at −18 °C. This procedure was repeated until a wet weight of feces of 80 g was achieved.

At the end of the sampling, feces were freeze-dried and homogenized, followed by separation of the samples for analysis of yttrium, crude protein, dry matter and lipids. The apparent digestibility coefficient (ADCn) for protein, dry matter and lipids was calculated as
$$\%\;\mathrm{ADCn}=100-[100\times \left(\frac{\%\;C\;\mathrm{diet}}{\%\;C\;\mathrm{faeces}}\right)\times \left(\frac{\%\;N\;\mathrm{faeces}}{\%\;\mathrm{Ndiet}}\right)],$$
where C is the yttrium value and N is the concentration of the nutrient considered (in % of dry matter).

ADC for dry matter, protein, and lipids calculations was calculated as
$$\%\mathrm{ADCingr}=\mathrm{ADC}\;\mathrm{dieta}\;\mathrm{teste}+\left[\left(\mathrm{ADC}\;\mathrm{dieta}\;\mathrm{teste}-\mathrm{ADC}\;\mathrm{dieta}\;\mathrm{refer}\hat{e}\mathrm{ncia}\right)\times \frac{0.85\;X\;N\;\mathrm{ref}}{0.15\;X\;N\;\mathrm{ing}}\right],$$
where ADCingr is ADC calculated in the first equation, ingr is the ingredient concentration of what is being calculated (dry matter, protein, lipids, and fatty acids), and ref represents the values of what is being calculated in the reference diet.

#### 2.2.2. Feed Additive Assay

The experiment using the microalgae meal as feed additive consisted of feeding shrimp in a clear water system. The experiment was designed randomly with four replicate tanks in a total of 16 experimental units. The four different levels (0, 0.5, 1 and 2%) of *Nannochloropsis* spp. meal were tested. The experiment proceeded for two weeks (15 days) in a room containing a marine water distribution system, aeration (6.07 ± 0.38 mg L^−1^) and heating (28.75 ± 0.71 °C). The experimental units were polyethylene, with a flat bottom, and capacity of 500 L of water. 

All tanks were filled with marine water from Barra da Lagoa (Florianópolis, SC, Brasil) with a salinity of 31.74 g L^−1^, an alkalinity of 132.8 mg L^−1^, pH 8.00, ammonia 0.3 mg L^−1^ and nitrite 0 mg L^−1^. Each experimental unit was populated with 20 shrimp weighing an average of 6.05 ± 0.06 g. Feed was offered four times a day (08 h 30 min, 12 h, 14 h 30 min and 17 h) according to a feeding table. Water exchange was made once a day during the afternoon at 80% of tank volume with the removal of the excess of organic matter.

During the experiment, pH (8.06 ± 0.06), alkalinity (124.70 ± 3.20), salinity (33.40 ± 0.10), ammonia (<0.5 mg L^−1^), and nitrite (<0.14 mg L^−1^) were kept within the limits for marine shrimp rearing [36]. Alkalinity analysis followed the method of Alpha (1995) [37], and nitrite and ammonia analyses were performed according to Strickland and Parsons (1972) [38]. Dissolved oxygen, temperature, pH and salinity were measured using the YSI—Professional Plus handheld multiparameter meter. 

### 2.3. Analysis of Diets

Analysis of the diets followed AOAC (Association of Official Analytical Chemists, 1999) [39] methodology. Diets were submitted to analyses of dry matter (drying at 105 °C to constant weight, method 950.01), crude protein (Kjeldahl, method 945.01), total lipid (Soxhlet, method 920.39C), ash (incineration at 550 °C, method 942.05) and gross energy (adiabatic bomb calorimeter). Fatty acid analyses were measured by gas chromatography using the modified method of Folch, Less, and Stanley (1957) [40], as described by Corrêa, Nobrega, Mattioni, and Fracalossi (2018) [41].

### 2.4. Midgut Microbiology Analysis

At the end of the experiment, the digestive tracts of five shrimp per tank (twenty animals per treatment) were collected. The digestive tracts were aseptically extracted, homogenized in a grinder, serially diluted (1/10) in sterile 3% saline solution, and sown in culture medium (Agar Marine and Agar Thiosulfate Citrate Bile Sucrose—TCBS) for counting total Vibrionaceae and heterotrophic bacteria, respectively. After 24 h of incubation at 30 °C, the total colony-forming units (CFU) were counted.

### 2.5. Thermal Shock Trial

Ten shrimp from each treatment were transferred from tanks with seawater at 28.75 ± 0.71 °C to a 60 L aquarium filled with seawater at 12.0 ± 0.3 °C under constant aeration and kept in these conditions for 1 h. Afterwards, they were transferred back to the tanks with seawater at 28.75 ± 0.71 °C, and survival was monitored for 48 h post-thermal shock (hpts). The seawater used in all thermal shock trials was from the same reservoir, which maintained salinity at 33.30‰.

### 2.6. Hemato-Immunological Analysis

At the end of the experiment, three animals from each replicate (twelve animals per treatment) were submitted to hemolymph collection from the ventral part of the body using 1 mL sterile syringes with a 21G needle and cooled at 4 °C. Forty microliters of hemolymph from each animal were fixed in a solution of 4% formaldehyde/modified Alsever’s solution (MAS) (sodium citrate 27 mM; EDTA 9 mM; glucose 115 mM; NaCl 336 mM; pH 7.0) for hemocyte counting. The remaining hemolymph was left to clot in ice for 2 h and then macerated and centrifuged at 10,000× *g* for 10 min to obtain the serum, which was then aliquoted and stored at −20 °C for later use in other immunological analyses. The number of hemocytes per milliliter of hemolymph was estimated by direct counting in a Neubauer chamber with phase-contrast microscopy [42].

Phenoloxidase activity (PO) was determined with spectrophotometry (DO 490 nm) through the formation of dopachrome pigment after the oxidation of the substrate l-3,4-dihydroxyphenylalanine (l-DOPA, Sigma Chemical Co., Saint Louis, MO, USA), using the methodology described by Söderhäll and Häll (1984) [43], and performed in triplicate. Serum samples were diluted (1:7) in TBS-2 (10 mM Tris, 336 mM NaCl, 5 mM CaCl_2_, 10 mM MgCl_2_, pH 7.6), and 50 μL of this solution was preincubated with an equal volume of the enzyme trypsin (Sigma, 1 mg mL^−1^) for 15 min at 20 °C in a 96 microwell plate (flat bottomed). In control, trypsin and serum were replaced by TBS-2. After incubation, 50 μL of l-DOPA (3 mg mL^−1^) was added to the wells, and dopachrome formation was monitored after 5 and 10 min. One unit of specific activity from PO was equivalent to the variation of 0.001 in the absorbance min^−1^ mg^−1^ protein. Protein concentration in the hemolymph was estimated by a method described by Bradford (1976) [44], using bovine serum albumin (BSA) as the standard. 

To determine the serum agglutination titer, 50 μL shrimp serum samples were serially diluted in TBS-1 (50 mM Tris, 150 mM NaCl, 10 mM CaCl_2_, 5 mM MgCl_2_, pH 7.4) in a 96-microwell plate with concave bottom. Each well received 50 μL of 2% solution of dog erythrocytes in TBS-1, and plates were incubated for 2 h at room temperature in a humidified chamber. The control was prepared through the substitution of serum by TBS-1, and all analysis was done in triplicate. Serum agglutination titer was defined as the reciprocal value obtained from the highest dilution capable of agglutinating erythrocytes, according to Maggioni et al. (2004) [45].

For the analysis of reactive oxygen species (ROS), three animals from each tank were used for the collection of hemolymph in the pre-shock period (hour 0), 1 h after the shock and 24 h after shock. The production of superoxide anion (O_2_^−^) by hemocytes was measured using the NBT reduction method (nitro-blue-tetrazolium) according to Guertler et al. (2010) [2]. As a cellular activator, laminarin (β-1.3 glycan) (Sigma-Aldrich) was used. Analyses were performed in triplicate.

### 2.7. Data Analysis

Bacterial count data were transformed into log_10_ (x + 1), and serum agglutination/antimicrobial titer data were transformed into log_2_ (x + 1) before being subjected to statistical analysis. Data homoscedasticity and normality were assessed with the Levene test and Shapiro–Wilk test, respectively. Bacteria count and hemato-immunological parameters were subjected to unifactorial variance analysis (ANOVA one way), followed by the Tukey test. Thermal shock survival data were analyzed by Kaplan–Meier. All statistical tests were evaluated with a significance level of 5% and were performed in Statistica 10 (StatSoft). 

## 3. Results

The shrimp finished the experiment weighing 9.17 ± 0.37 g, with weekly growth of 1.56 g, and no mortality was observed during the experimental period.

### 3.1. Digestibility of Nannochloropsis spp. Meal

In general, the apparent digestibility coefficient (ADC) of *Nannochloropsis* spp. meal was high (Table 4). The ACD from longchain polyunsaturated fatty acids was 100%, and the fatty acids belonging to the n-3 group reached 95%. It was possible to observe the complete digestibility of some fatty acids such as docosapentaenoic, arachidonic, linolenic, linoleic and oleic, as well as protein. A high digestibility of total lipids (78.88%) and eicosapentaenoic fatty acid (73.86%) was also observed.

### 3.2. Midgut Microbiology 

No significant difference (*p* ≥ 0.05) was noted in total heterotrophic bacteria and Vibrionaceae in the intestine of *Litopenaeus vannamei* shrimp fed for 15 days with diets containing 0, 0.5, 1 and 2% of *Nannochloropsis* spp. meal (Figure 1). 

### 3.3. Thermal Shock Resistance 

The results for thermal shock resistance show significant differences in all treatments when compared to the control group (Figure 2), except the treatment with 1% of inclusion. Therefore, the inclusion of *Nannochloropsis* spp. in diets for marine shrimp promoted resistance to the thermal shock, such that shrimp fed with diets containing the microalgae presented lower mortality when compared to the control group.

### 3.4. Immunology

No statistical difference was observed in total hemocyte counts, protein concentration, phenol-oxidase activity and agglutination titer among the treatments (*p* ≥ 0.05) for *Litopenaeus vannamei* fed for 15 days with diets containing 0, 0.5, 1 and 2% of *Nannochloropsis* spp. (Table 5).

The results of reactive oxygen species (ROS) reveal a higher production in shrimp fed with 2% of *Nannochloropsis* spp. microalgae when compared to other groups at 0, 1, and 24 h after thermal shock (Table 6). 

Regarding the induction capacity of hemocytes after laminarin stimulation (Figure 3), a difference in O_2_^−^ production in the treatment groups fed with 1 and 2% of *Nannochloropsis* spp. was observed. Animals fed the diet containing 1% of microalgae presented around 2.5 times more ROS production compared to values in basal conditions. On the other hand, shrimp fed diets with 2% microalgae presented the opposite behavior, since the production of ROS was 2.6 times higher before induction with laminarin.

## 4. Discussion

*Nannochloropsis* spp. are microorganisms with the capacity to produce many beneficial compounds for animals. For instance, they have a high capacity to produce fatty acids (mainly those from the n-3 series), and they present immunostimulants, such as b-glucans, which can interact with the immune system, in either vertebrates or invertebrates [29,46]. The benefits of *Nannochloropsis* spp. in shrimp were demonstrated in some works as increasing growth performance, survival and stimulation of the immune system. Thus, in the present work, the effects *Nannochloropsis* spp. As a feed additive for Pacific white shrimp were assessed. 

Digestibility of the microalgae meal was tested, and it was high. In general, protein digestibility, as well as that of some very important fatty acids, such as arachidonic, linoleic, linolenic and eicosapentaenoic, was respectively high. The digestibility of lipids and the eicosapentaenoic fatty acid was also high. This result is very important because it demonstrates that these shrimp can utilize nutrients present in the microalgae meal. Therefore, it can be concluded that microalgae meal is an ingredient highly assimilable by animals. Accordingly, it is likely that such connections are due to the fact that this feeding source is already part of the food chain of shrimp in the natural environment. 

The addition of microalgae meal in diets did not interfere with bacterial midgut microbiology of the intestinal tract; nor did it alter some immunological parameters, such as total hemocyte counts, protein concentration, phenol-oxidase activity, and serum agglutination titer. However, statistical differences were found in the thermal shock challenge and reactive oxygen species (ROS) production. 

Thermal shock challenge provided the largest set of significant differences in the present study. The inclusion of *Nannochloropsis* spp. in the diets of shrimp induced thermal shock resistance, as shown by the lower mortality of groups with 0.5% and 2% inclusion when compared to control groups. Even the treatment 1% did not show a significant difference from control; the mortality was lower, and the p value was 0.064—therefore, this was close to the significant difference. This finding has already been observed in shrimps after being fed with *Sargassum filipendula* macroalgae [47,48], where resistance to thermal shock from shrimp was also observed, as in the present study, with also showed *Nannochloropsis* spp. to be an immunostimulant which strengthened the immune system.

Marine shrimp are ectothermic organisms with no control over internal temperature; therefore, they are highly sensitive to temperature shifts in the rearing environment. Thus, low temperatures may lead to physiological changes, such as membrane fluidity loss, protein integrity loss, and oxidative stress [49,50]. Some studies have shown that some mechanisms of shrimp to regulate low temperature conditions, such as the regulation of lipid metabolism and modification of ionic transport, are impaired by the reduction in membrane fluidity, which is also very harmful to cellular functioning, as it leads to problems with cellular functioning and immobilizes transmembrane proteins [49,51,52,53]. Many biochemical mechanisms can increase the fluidity of the membrane, preventing its harmful effects. Among these mechanisms, it is known that the increase in fatty acids and cholesterol, unsaturated fatty acids and the restructuration their respective polar groups, as well as the elevation of long chain fatty acids, increase the fluidity of the membrane [53]. It has been described in the literature that unsaturated fatty acids are incorporated in cell membrane phospholipids in situations of cold exposition, leading to modifications in membrane fluidity. This mechanism is common in animals that have resistance to thermal variations, such as insects, crustaceans, plants, and microorganisms [49,50,53,54,55,56]. Thus, the data of the present study demonstrate that the presence of unsaturated fatty acids of n-3 series in *Nannochloropsis* spp. microalgae added in the diet could have been metabolized and helped to maintain membrane fluidity. The resistance to the thermal shock, as determined in this study, seems to prove this theory, as well as the high digestibility of fatty acids. Similar results were also demonstrated by Schleder et al. (2017) [48] through MALDI-TOF MS analysis, where the inclusion of *Sargassum filipendula* in the diet led to the increase in membrane fluidity and antimicrobial defense, also changing energetic and hemocyte lipid metabolism. 

In the present work, changes in the immune status of animals could be measured by evaluating reactive oxygen species (ROS) in shrimp hemocytes. Before thermal shock, we saw an increase of about two-fold (*p* ≤ 0.05) in the production of superoxide anion in the 2% treatment when compared to other groups. It is known that superoxide anion production is the first component produced in the ROS cascade during respiratory burst; therefore, this activation is more commonly observed during phagocytosis. Many studies suggest that b-glucans are immunostimulants and help in the prevention of disease in crustaceans [24,57], either by in vivo immune stimulation (for immersion and by feed additives) or in vitro, generating cellular responses through, for instance superoxide anions production [24,27]. In the present study, a crescent level of immunological stimulus was observed among treatments with 0, 0.5 and 1% of microalgae inclusion. On the other hand, animals fed 2% inclusion presented higher levels of immune stimulation, or ROS. This finding corroborates other studies demonstrating the increase in ROS in immunostimulated shrimp [24], in shrimp infected by viruses, such as Taura syndrome [58], white spot virus and hypodermal infection and hematopoietic necrosis [25,59].

Increased ROS production was also seen 1 h after thermal shock in all experimental groups. This increase could have been caused by hypoxia and reoxygenation that happens in cases of a severe reduction in metabolism during thermal shock and an abrupt return to normal temperature. Some studies have demonstrated that this situation promotes ROS production in the organism [60,61,62], as well as instances of high oxygen saturation [4]. At 24 h after thermal shock, ROS production was again similar to the that found before the challenge, demonstrating that the animals could return to basal levels. This is very important, because the presence of toxic molecules, such as ROS, in the absence of infection represents energetic overload for the organism and can lead to tissue damage [63].

After laminarin stimulation, an increase of 2.5-fold in ROS production was observed in hemocytes in the group fed 1% of microalgae 24 h after thermal shock when compared to basal values. This result indicates that this microalgae concentration in the diet could enhance the capacity of hemocytes to produce ROS after stimulation. On the other hand, during the same 24 h period, the shrimp fed 2% microalgae presented a high basal production in which hemocytes decreased ROS production in response to laminarin by 2.6-fold. These results suggest that hemocytes of these animals were already at respiratory burst, even before in vitro induction. It is also possible that the cells of these animals were unable to generate ROS, indicating a possible low immunological competence of these animals during this period. Thus, even though shrimp fed 2% diets had higher ROS values at basal level, this was not accompanied by an increase in hemocyte capacity in producing ROS after laminarin stimulation, as evidenced in the group fed the 1% diet.

No statistical differences were found in some immune parameters, but it was still possible to observe a tendency of the increase in microalgae inclusion in diets and a further decrease at 2% inclusion. This fact suggests an immune stimulation of the animals that can be observed by the decrease in agglutination titer, which is caused by an alteration in stress and infection, as widely reported in the literature. These data corroborate the studies of Maggioni, Andreatta, Hermes, and Barracco (2004) [45] who demonstrated the same decrease in agglutination titer after unilateral ablation in adult female *Litopenaeus vannamei.* Furthermore, the decrease in coagulation capacity of the hemolymph and drop of total hemocyte counts could be associated with infection, stress or immunostimulation [6,24,59]. Under such conditions, the hemocytes migrate and infiltrate the infected tissues [64]. This event also happens in crustaceans infected with white spot syndrome virus and infectious myonecrosis virus [2,59,65,66]. 

It is important to highlight that phenol-oxidase (PO) activity increased in shrimp fed with 2% microalgae and began to decrease later in the experiment. This increase is important because the PO activity system acts in diverse defense situations and immune stimulation, such as phagocytosis, encapsulation and nodule formation which ends in melanized tissue [28,59]. This decrease in PO activity and protein concentration corroborates studies that demonstrate shrimp in stress conditions. The initial increase in PO and its following decrease suggest partial suppression of proPO that is likely caused by stress [6,27,65].

The treatments where shrimp were fed diets of 0.5% and 1% inclusion seem to better prepare shrimp against an eventual infection and without much energetic cost, when compared to 2% inclusion. Some authors agree that the increase in total hemocyte count (CTH) provides better protection of shrimp against infections since hemocytes are mainly responsible for immune cellular reactions and are the site of many defense molecules’ expression in the organism. It also has better protection, with an increase in the capacity of hemolymph agglutination, regulation of ROS and the hemocyte capacity of production of ROS after laminarin stimulation [67,68,69].

Researchers believe that having an immune system constantly activated is not beneficial for animals; however, strengthening the immune system, seeking higher immune competence, and, consequently, improving the capacity to fight off infection are all worth pursuing [63]. Accordingly, a positive outcome of this study is the fact that feeding microalgae for only 15 days led to immune stimulation the shrimp. This result is important because, within a short period of time, it is possible to induce immunostimulation in shrimp though simple feeding and prepare them for possible stress events, such as weather changes, thus avoiding disease outbreaks. 

Future studies might test the influence of microalgae inclusion on zootechnical parameters of shrimp and also challenge shrimp with bacteria of the genus *Vibrio* or white spot syndrome virus to assess if immune-stimulated shrimp respond better after infection.

## 5. Conclusions

The digestibility of *Nannochloropsis* spp. as seen in *Litopenaeus vannamei* was high for all nutrients. The inclusion of 0.5 and 2% of *Nannochloropsis* spp. in the diet increased the resistance of shrimp during thermal stress and did not interfere with the midgut microbiology.

The hemocytes of animals fed 1% of microalgae in their diet were effectively able to increase ROS and superoxide production after in vitro stimulation by laminarin and presented a better return to basal levels. Animals fed 2% inclusion seem to be more immune-stimulated at basal level, since they could produce more ROS, which could be used in the event of infection.

## Figures and Tables

**Figure 1 animals-11-00150-f001:**
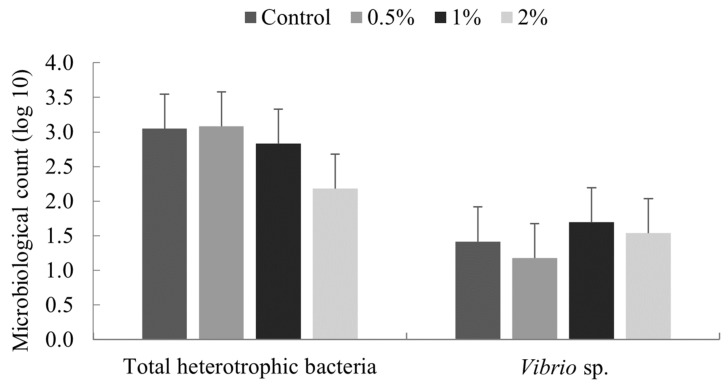
Colony counts of bacteria from the intestine of *Litopenaeus vannamei* fed for 15 days with diets containing 0, 0.5, 1 and 2% of *Nannochloropsis* spp. meal. Scale bar indicates the standard deviation of the average (*n* = 4).

**Figure 2 animals-11-00150-f002:**
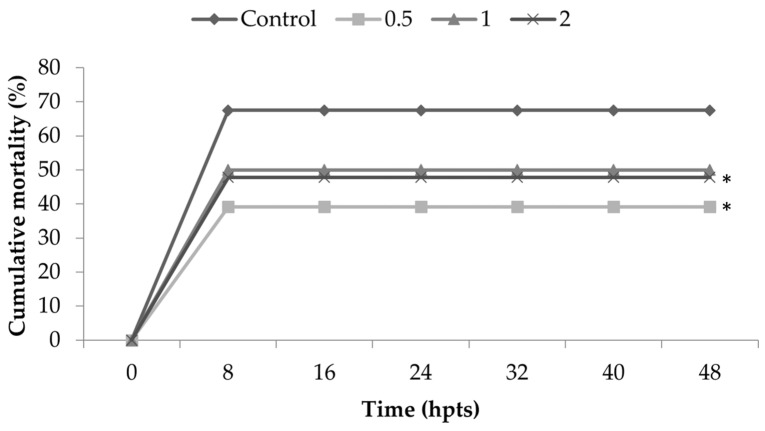
Cumulative mortality of *Litopenaeus vannamei* fed for 15 days with diets containing 0.5, 1 and 2% of *Nannochloropsis* spp. and control diet (0% inclusion). Shrimp were observed for 48 h post-thermal shock (hpts). Statistical differences among the treatments were observed (*p* = 0.03633). In comparison with the control (0%), the statistical differences were as follows: 0.5% group (*p* = 0.00446) and 2% group (*p* = 0.03898). The group fed 1% presented no statistical difference (*p* = 0.06412). * corresponds to statistical difference in comparison to control.

**Figure 3 animals-11-00150-f003:**
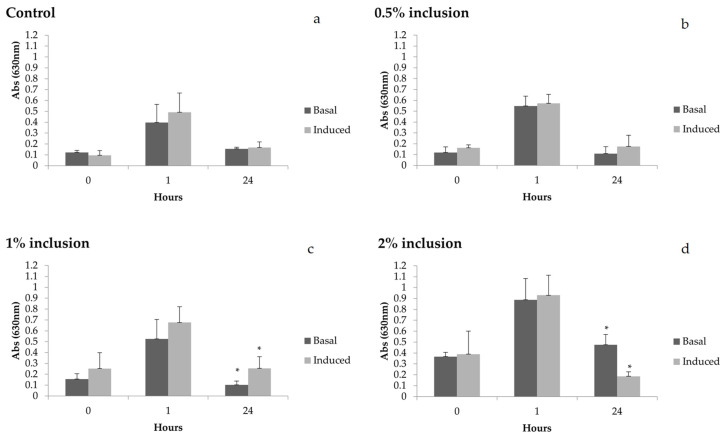
Reactive oxygen species (ROS) in basal and induced (laminarin) conditions before thermal shock (0 h) and after thermal shock (1 and 24 h) in *Litopenaeus vannamei* fed diets containing 0 (**a**), 0.5 (**b**), 1 (**c**) and 2% (**d**) of *Nannochloropsis* sp inclusion for 15 days. Scale bar represents average standard deviation (n = 4). Hours with * indicate statistical difference between basal and induction conditions. * Statistical difference found in the production of superoxide anions between basal and induced levels, in the treatment with 1% inclusion of the microalgae, in collection 24 h after the thermal shock (*p* = 0.03639), and in collection 24 h after the thermal shock in the treatment with 2% inclusion (*p* = 0.00138).

**Table 1 animals-11-00150-t001:** Proximate composition and nutritional information of *Nannochloropsis* spp. meal, including the most representative fatty acids.

Nutrient	Composition (g 100 g^−1^ Dry Matter) ^a^
Dry matter	97.16
Protein	42.85
Lipids	18.95
Fatty acids	
14:00	0.59
16:00	2.85
18:1 n-9	0.88
20:4 n-6	0.84
20:5 n-3	2.97
SFA ^b^	4.60
MUFA	4.08
PUFA	4.55
PUFA n-6	1.38
PUFA n-3	3.07

^a^ Analysis made in the Aquatic Species Nutrition Laboratory described in Section 2.3. ^b^ Fatty acid groups: SFA = saturated; MUFA = monounsaturated; PUFA = polyunsaturated.

**Table 2 animals-11-00150-t002:** Diet formulation for the digestibility trial.

Ingredient (g kg^−1^ Dry Weigh)	Reference Diet	Test Diet: *Nannochloropsis* spp.
Casein ^a^	256.30	230.67
Corn starch ^a^	223.90	201.51
Gelatine ^a^	170.30	153.27
*Nannochloropsis* sp meal ^b^	0.00	100.00
Kaolin ^a^	114.00	102.60
Cellulose ^a^	53.04	47.74
Lecitin	50.12	45.11
Magnesium sulfate	25.20	22.68
Cod liver oil	20.00	18.00
Potassium chloride	16.40	14.76
Mineral Premix ^c^	16.20	14.58
Soy oil	16.00	14.40
Sodium chloride	14.40	12.96
Mono calcium phosphate	13.30	11.97
Carboxymethylcellulose ^a^	4.24	3.82
Vitamin Premix ^c^	3.80	3.42
Choline hydrochloride	1.00	0.90
Yttrium oxide	1.00	0.90
Vitamin C ^d^	0.80	0.72
Composition (g 100 g^−1^ dry matter) ^e^		
Dry matter	93.65	94.22
Protein	41.07	41.17
Lipids	7.48	8.96
SFA ^f^	1.48	2.33
MUFA	1.83	2.33
PUFA	3.71	3.69
PUFA n-6	2.29	2.50
PUFA n-3	0.88	1.19
n-3/n-6	0.39	0.48

^a^ Distributed by Rhoster (Araçoiaba da Serra, São Paulo, Brasil). ^b^ Produced by Necton—PhytoBloom (Algarve, Portugal). ^c^ Composition of Vitamin and Mineral premix distributed by In Vivo Nutrição e Saúde Animal Ltd.a. (São Paulo, SP, Brasil): vit. A—900 mg kg^−1^; vit. D3—25 mg kg^−1^; vit. E—46,900 mg kg^−1^; vit. K3—1400 mg kg^−1^; vit. B12—50 mg kg^−1^; Vit. B6—33,000 mg kg^−1^; riboflavin—20,000 mg kg^−1^; nicotinic acid—70,000 mg kg^−1^; pantothenic acid—40,000 mg kg^−1^; biotin—750 mg kg^−1^; folic acid—3000 mg kg^−1^; copper—2330 mg kg^−1^, zinc—10,000 mg kg^−1^; manganese—6500 mg kg^−1^; selenium—125 mg kg^−1^; iodine—1000 mg kg^−1^; cobalt—50 mg kg^−1^; magnesium—20 g kg^−1^ and potassium—6.1 g kg^−1^. ^d^ L-ascorbic acid-2-monophosphate 35%. DSM Produtos Nutricionais Brasil (São Paulo, SP, Brazil). ^e^ Analysis performed by the Laboratory of Aquatic Species Nutrition, as described in item 2.3. ^f^ Fatty acid groups: SFA = saturated, MUFA = monounsaturated, PUFA = polyunsaturated.

**Table 3 animals-11-00150-t003:** Formulation and composition of the experimental diets for *Litopenaeus vannamei* containing microalgae (*Nannochlorosis* spp.) meal at different concentrations.

Ingredients (g kg^−1^ Dry Matter)	Microalgae Meal Inclusion (%)			
	0	0.5	1	2
Soybean meal ^a^	324.60	323.00	321.40	318.00
Wheat meal ^b^	150.00	149.00	148.20	147.00
Poultry meal ^c^	125.70	125.10	124.40	123.20
Fish meal ^d^	150.00	149.30	148.50	147.00
Soy lecitin	25.00	24.90	24.80	24.50
Soy oil ^e^	10.00	10.00	9.90	9.80
Fish oil ^f^	20.00	19.90	19.80	19.60
Microalgae meal (*Nannochloropsis* spp.)	0.00	5.00	10.00	20.00
Vitamin Premix ^g^	5.00	5.00	5.00	4.90
Mineral Premix ^g^	17.00	16.90	16.80	16.70
Monocalcium phosphate	25.00	24.90	24.80	24.50
Carboxymethylcellulose	5.00	5.00	5.00	4.90
Potassium chloride	10.00	10.00	9.90	9.80
Methionine	5.00	5.00	5.00	4.90
Vitamin C ^h^	0.70	0.70	0.70	0.70
Magnesium sulfate	15.00	14.90	14.90	14.70
Sodium chloride	12.00	11.90	11.90	11.80
Kaolin	100.00	99.50	99.00	98.00
Composition (g 100 g^−1^ dry matter)				
Dry matter	97.80	97.30	96.80	95.80
Protein	35.86	35.68	35.50	35.14
Lipids	10.00	9.95	9.90	9.80
Ash	18.28	18.19	18.10	17.91
Crude energy (kcal kg^−1^)	3,762.00	3,743.30	3,724.40	3,686.80

^a^ Soybean meal distributed by BRF Ingredients (Itajaí, Santa Catarina, Brasil): 56.61 mg kg^−1^ crude protein and 4500 kcal kg^−1^ crude energy. ^b^ Wheat meal: Rosa Branca, Type 1: 13 mg kg^−1^ crude protein and 3440 kcal kg^−1^ crude energy. ^c^ Poultry meal distributed by BRF Ingredients (Itajaí, Santa Catarina, Brasil): 62 mg kg^−1^ crude protein and 4661 kcal kg^−1^ crude energy. ^d^ Fish meal distributed by Agroforte Ind (Biguaçu, Santa Catarina, Brasil): 59.35 mg kg^−1^ crude protein and 4056.13 kcal kg^−1^ crude energy. ^e^ Soy oil: Soya: 7909 kcal kg^−1^ crude energy. ^f^ Fish oil distributed by BFP Bio Food Products (Itajaí, Santa Catarina, Brasil): 8445 kcal kg^−1^ crude energy. ^g^ Composition of Vitamin and Mineral premix distributed by In Vivo Nutrição e Saúde Animal Ltd.a. (São Paulo, SP, Brazil): vit. A—900 mg kg^−1^; vit. D3—25 mg kg^−1^; vit. E—46,900 mg kg^−1^; vit. K3—1400 mg kg^−1^; vit. B12—50 mg kg^−1^; Vit. B6—33,000 mg kg^−1^; riboflavin—20,000 mg kg^−1^; nicotinic acid—70,000 mg kg^−1^; pantothenic acid—40,000 mg kg^−1^; biotin—750 mg kg^−1^; folic acid—3000 mg kg^−1^; copper—2330 mg kg^−1^; zinc—10,000 mg kg^−1^; manganese—6500 mg kg^−1^; selenium—125 mg kg^−1^; iodine—1000 mg kg^−1^; cobalt—50 mg kg^−1^; magnesium—20 g kg^−1^ and potassium—6.1 g kg^−1^. ^h^ L-ascorbic acid-2-monophosphate 35%. DSM Produtos Nutricionais Brasil (São Paulo, SP, Brazil).

**Table 4 animals-11-00150-t004:** Diet composition and apparent digestibility coefficient (ADC) of *Nannochloropsis* spp. meal for the Pacific white shrimp.

Nutrient ^a^	Diet with *Nannochloropsis* spp. Composition g 100 g ^−1^ Dry Matter	ADC %
Dry matter	97.16	95.43 ± 5.25
Protein	18.95	100.00 ± 2.59
Lipids	42.85	78.77 ± 3.11
14:0 Miristic	0.15	85.43 ± 3.96
16:0 Palmitic	1.23	94.18 ± 5.09
18:0 Stearic	0.24	100.00 ± 6.32
18:1 n-9 Oleic	1.32	100.00 ± 6.18
18:2 n-6 Linoleic	2.40	100.00 ± 6.50
18:3 n-3 Linolenic	0.16	100.00 ± 6.07
20:4 n-6 Araquidonic	0.09	100.00 ± 2.05
20:5 n-3 EPA	0.41	73.86 ± 2.48
22:6 n-3 DHA	0.19	100.00 ± 6.51
SFA ^b^	2.33	90.42 ± 4.60
MUFA	2.33	97.51 ± 5.51
PUFA	3.69	100.00 ± 5.98
PUFA n-6	2.50	100.00 ± 6.33
PUFA n-3	1.19	95.44 ± 5.25

^a^ Value expressed as average n = 3 followed by standard deviation. ^b^ Fatty acids: SFA = saturated, MUFA = monosaturated, PUFA = polyunsaturated.

**Table 5 animals-11-00150-t005:** Immunological parameters (hemocyte counts, protein concentration, phenol-oxidase activity and agglutination titer) of *Litopenaeus vannamei* fed for 15 days with diets containing 0, 0.5, 1 and 2% of *Nannochloropsis* spp.

Treatments	Total Hemocyte Count (10^6^ Cells mL^−1^)	Protein Concentration (mg mL^−1^)	PO Activity (Unit min^−1^ mg^−1^ Protein)	Agglutination Titer (log_2_ x + 1)
Control	28.39 ± 1.90	517.54 ± 1.22	28.90 ± 1.00	8.35 ± 0.47
0.5%	30.19 ± 2.83	516.48 ± 0.31	36.10 ± 0.56	8.25 ± 1.50
1%	32.75 ± 4.27	516.44 ± 0.71	35.00 ± 1.44	8.50 ± 1.29
2%	27.84 ± 1.25	516.28 ± 1.03	26.30 ± 0.60	8.00 ± 0.82
p	0.1879	0.2212	0.4424	0.9292

These data represent average ± standard deviation of *n* = 4. No statistical difference was observed among treatments (*p* ≥ 0.05).

**Table 6 animals-11-00150-t006:** Reactive oxygen species results of *Litopenaeus vannamei* fed for 15 days with diets containing 0, 0.5, 1 and 2% of *Nannochloropsis* spp.

Treatments	Reactive Oxygen Species (ROS)—Basal		
	0 Hora *	1 Hora	24 Horas
Control (0%)	0.13 ± 0.02 ^a^	0.40 ± 0.17 ^a^	0.15 ± 0.01 ^a^
0.5%	0.12 ± 0.05 ^a^	0.55 ± 0.09 ^a^	0.11 ± 0.06 ^a^
1%	0.16 ± 0.05 ^a^	0.53 ± 0.18^a^	0.10 ± 0.04 ^a^
2%	0.37 ± 0.04 ^b^	0.89 ± 0.20 ^b^	0.47 ± 0.09 ^b^

* Sampling before thermal shock (0 h) and then 1 and 24 h after thermal shock. Data represent average ± standard deviation (n = 3). Different letters in the same column represent statistical differences according to the Tukey test (*p* < 0.05).

## Data Availability

Data available on request from the authors.

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
