# Peer review of "Nannochloropsis spp. as Feed Additive for the Pacific White Shrimp: Effect on Midgut Microbiology, Thermal Shock Resistance and Immunology"

_animals, 2021, doi:10.3390/ani11010150_

Round 1

Reviewer 1 Report

It is a well-organized and written article and I detected just little mistakes that should be corrected easily.

Abstract:

Line 25, the word “and” must be included in the phrase “thermal shock resistance and immunological parameters”

Materials and Methods:

Line 90, write in italics letter L. and Penaeus monodon.

In table 2, decide

In tables 2 and 3 correct Nannochlorosis, and change the capital letter in Premix from table 2.

Consult your style guide if you are uncertain of using a comma before coordinating conjunction and the final item in a series, like on page 6 line 136 (8h, 12h, 14h, and 18h) or not, as on page one line 29 (0, 0.5, 1 and 2% inclusion).

Page 6 line 143 Pausteur must be corrected.

Clarify a little more the equations of lines 151 to 157.

In the same paragraph, you are using different annotations types (mg L-1) and (mg. L-1), please check the whole paper.

Change the semicolon by a comma on line 168.

Check out the whole paper regarding the use of °C. Compare these on page 7. 

Check out the whole paper regarding the references and text of the paper, page 8 line 228.

Results:

Use italics for Vibrio in figure 1. Page 10.

If you are using 0.5 along with the text, why in line 300 you use 0,5? Legend of figure 3.

Discussion:

No discussion was mentioned as to why there were significant differences between the control and concentrations of 0.5 and 2% inclusion, but not for 1%.

Author Response

RV1: Abstract: Line 25, the word “and” must be included in the phrase “thermal shock resistance and immunological parameters”

Answer: It  was modified.

RV1: Materials and Methods: Line 90, write in italics letter L. and Penaeus monodon.

Answer: It  was modified.

RV1: In table 2, decide

Sorry, I did not understand.

RV1: In tables 2 and 3 correct Nannochlorosis, and change the capital letter in Premix from table 2.

Answer: Adjusted Nannochlorosis for Nannochloropsis in table 2 and 3, and capital letter in Premix from table 2.

RV1: Consult your style guide if you are uncertain of using a comma before coordinating conjunction and the final item in a series, like on page 6 line 136 (8h, 12h, 14h, and 18h) or not, as on page one line 29 (0, 0.5, 1 and 2% inclusion).

Answer: it was modified, removing the comma.

RV1:Page 6 line 143 Pausteur must be corrected.

Answer: I was corrected.

RV1: Clarify a little more the equations of lines 151 to 157.

Answer: I was modified, detailing the equation further.

RV1: In the same paragraph, you are using different annotations types (mg L-1) and (mg. L-1), please check the whole paper.

Answer: I was modified. Whole article revised and changed to (mg L-1).

RV1: Change the semicolon by a comma on line 168.

Answer: I was changed.

RV1: Check out the whole paper regarding the use of °C. Compare these on page 7. 

Answer: I was modified. The space between the number and the ° C was removed.

RV1: Check out the whole paper regarding the references and text of the paper, page 8 line 228.

Answer: I was modified. Every reference has been revised.

RV1: Results:Use italics for Vibrio in figure 1. Page 10.

Answer: I was modified

RV1: If you are using 0.5 along with the text, why in line 300 you use 0,5? Legend of figure 3.

Change made: Changed to comma.

RV1: Discussion: No discussion was mentioned as to why there were significant differences between the control and concentrations of 0.5 and 2% inclusion, but not for 1%.

Answer: We added one sentence to the discission. The 1% treatment was not significative different; however, the p-value was 0,064. The difference between 0.5% and 1% survival was only 1 animal. Probably, with the number o animal used was higher, we would have seen a significant difference.

Reviewer 2 Report

Nannochloropsis spp. as feed additive for the Pacific white shrimp: effect on intestinal microbiota, thermal shock resistance and immunology is a good title reflecting the content of the paper and provides some innovative ideas and tests a well known algal supplement in terms of its functionality properties. It demonstrates the potential for the most important shrimp species globally and merits attention.

The methodology is good and is appropriate to the topic in question as we need viable alternative solutions.

The parameters for immune function and the antioxidant status is very intercutting an you have clear evidence for benefits of the test ingredient on the shrimp defence system.  Your output of data is transparent and supports you hypothesis well.

My only issue is that the term microbiome is quite misleading and gives a wrong direction. Microbiome these days is fashionable to state in such nutrition papers but you have not undertaken this using full genomic sequencing for 16S RNA DNA gene expression work. Use new subheading to reflect classical microbial count measurements as actually undertaken as comprehensive profiling of the microbial gut ecology in this species was not done. You should re- phrase this expression with a new title and better terminology for the microbiology work. Its too basis to really call it the microbiome. 

The authors should also provide much more information on the various feed ingredients in the table for feed formulation as this is not in sufficient detail. Such materials as fishmeal and soybean meal must be given further details of their sourcing wherever possible.

In the main you provide a well written manuscript that is succinct and would be of interest to the shrimp farming and wider aquaculture industry. This a worthy and interesting investigation.

Author Response

RV2: My only issue is that the term microbiome is quite misleading and gives a wrong direction. Microbiome these days is fashionable to state in such nutrition papers but you have not undertaken this using full genomic sequencing for 16S RNA DNA gene expression work. Use new subheading to reflect classical microbial count measurements as actually undertaken as comprehensive profiling of the microbial gut ecology in this species was not done. You should re- phrase this expression with a new title and better terminology for the microbiology work. Its too basis to really call it the microbiome. 

Answer: I was modified. Microbiota term was changed to midgut microbiology, considered more appropriate according to other works already published.

RV2: The authors should also provide much more information on the various feed ingredients in the table for feed formulation as this is not in sufficient detail. Such materials as fishmeal and soybean meal must be given further details of their sourcing wherever possible.

Answer: I was modified. More information was included in table 3.